# Bit-mask Robust Contrastive Knowledge Distillation for Unsupervised Semantic Hashing

Submitted for Blind Review

## ABSTRACT

Unsupervised semantic hashing has emerged as an indispensable technique for fast image search, which aims to convert images into binary hash codes without relying on labels. Recent advancements in the field demonstrate that employing large-scale backbones (e.g., ViT) in unsupervised semantic hashing models can yield substantial improvements. However, the inference delay has become increasingly difficult to overlook. Knowledge distillation provides a means for practical model compression to alleviate this delay. Nevertheless, the prevailing knowledge distillation approaches are not explicitly designed for semantic hashing. They ignore the unique search paradigm of semantic hashing, the inherent necessities of the distillation process, and the property of hash codes. In this paper, we propose an innovative Bit-mask Robust Contrastive knowledge Distillation (BRCD) method, specifically devised for the distillation of semantic hashing models. To ensure the effectiveness of two kinds of search paradigms in the context of semantic hashing, BRCD first aligns the semantic spaces between the teacher and student models through a contrastive knowledge distillation objective. Additionally, to eliminate noisy augmentations and ensure robust optimization, a cluster-based method within the knowledge distillation process is introduced. Furthermore, through a bit-level analysis, we uncover the presence of redundancy bits resulting from the bit independence property. To mitigate these effects, we introduce a bit mask mechanism in our knowledge distillation objective. Finally, extensive experiments not only showcase the noteworthy performance of our BRCD method in comparison to other knowledge distillation methods but also substantiate the generality of our methods across diverse semantic hashing models and backbones.

## KEYWORDS

image retrieval; semantic hashing; knowledge distillation

## 1 INTRODUCTION

Content-based image retrieval is a crucial image search problem in which similar images are retrieved from a database given a query image [21]. It has been widely applied in many web applications (e.g., iStock [1]). In recent years, we have witnessed a significant surge in visual data on the Internet. To address the exponential growth of data volume and expensive annotating requirements in large-scale databases, unsupervised semantic hashing methods have played a pivotal role [26]. These methods aim to maintain the semantic similarity of images by deep convolutional neural networks (e.g., VGG [40]) without relying on labels, where the original high-dimensional embedding of the image is converted into a compact hash code for presentation. Taking advantage of using hash codes for efficient search, semantic hashing has achieved impressive results in large-scale image retrieval [50].

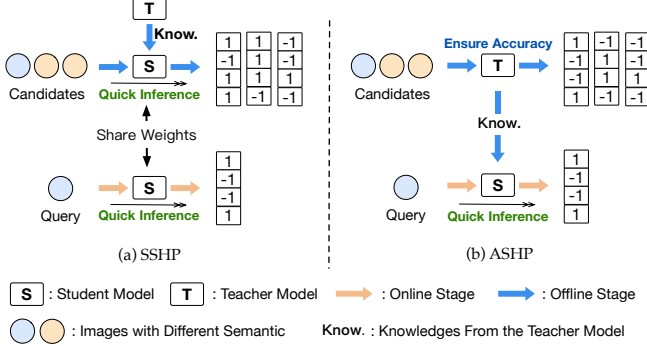

**Figure 1: Search paradigms for semantic hashing. (a) represents the Symmetric Semantic Hashing search Paradigm (SSHP), utilizing a single model for both offline and online stages. (b) depicts the Asymmetric Semantic Hashing search paradigm (ASHP), employing distinct models for each stage.**

With the emergence of more advanced architectures like Vision Transformer (ViT) [8], some pioneering studies [10, 19] have demonstrated their superiority in semantic hashing. For example, a recent publication [10] reports that by replacing the original backbone with ViT, the semantic hashing model CIBHash [34] gains around 46% improvement. This suggests a promising trend where semantic hashing approaches increasingly adopt such powerful backbones to extract meaningful representations. Unfortunately, a drawback arises when employing large-scale backbones. For example, we find that if we employ the large-scale model ViT_L_16 as the hashing model's backbone, it takes approximately 657 ms to process a batch of 64 images and generate hash codes. In contrast, the subsequent search process requires less than one-tenth of that time [2]. Consequently, the inference time becomes a bottleneck, adversely affecting the user experience in practical scenarios.

To promote the inference process, Knowledge Distillation (KD) is a frequently employed method [11], which compresses the large backbone of the teacher into a small student model for inference. The teacher tries to transform the necessary knowledge to the student for performance guarantee. Using such a technology, we have summarized two search paradigms for semantic hashing, including the symmetric and asymmetric search paradigms. First, as shown in Figure 1 (a), a conventional solution follows a symmetric search paradigm, and we name such a paradigm the Symmetric Semantic Hashing search Paradigm (SSHP). In SSHP, a single student model serves as the hash code extractor for semantically learning both candidates and queries in the offline and online stages, respectively. Additionally, another large teacher model takes charge of distilling the required knowledge to the student. Second, as shown in Figure 1 (b), we synthesize a novel paradigm named Asymmetric

---

[1] https://www.istockphoto.com/

[2] See Appendix F for details

Semantic Hashing search Paradigm (ASHP) for semantic hashing. The ASHP retains a powerful large teacher model offline to generate better hash codes of candidates for accuracy but leverages a lightweight student model online to facilitate quick search. This paradigm draws inspiration from some works [4, 23, 47, 56] that employ the methodology of firstly generating hash codes, followed by the learning of a hash function, which means we can deploy different models for the same hash code. The ASHP improves accuracy by sacrificing offline inference time while ensuring online inference efficiency for good user experiments.

At this point, we hope to grasp a suitable distillation method that can be used for both paradigms. However, the current knowledge distillation methods lack explicit design considerations for semantic hashing, thereby overlooking crucial factors inherent to semantic hashing. We identify three critical factors for designing an effective knowledge distillation strategy that facilitates the semantic hashing problem. (1) *Semantic space alignment.* In the symmetric paradigm (SSHP), the hash codes outputted from different stages are in the same space naturally because one single model is applied in both stages. However, in the asymmetric paradigm (ASHP), since we implement different models offline and online, the Hamming semantic space of both is extremely misaligned, which argues a necessary distillation way to ensure the space is consistent for effective search. (2) *Robust optimization.* Some current works [12, 15, 29, 41] target to ensure robustness in the hash model because noise significantly impacts the performance of hash codes. When employing a powerful teacher model, we cannot guarantee the accuracy for all samples, making some supervision signals noise and leading to incorrect optimization directions. Hence, we argue that a robust knowledge distillation process should also be achieved to avoid such situations. (3) *Hash code property.* Desirable hash codes have unique properties in distribution. For example, some semantic hashing models [7, 22, 34, 60] tend to generate "bit independence" hash codes, resulting in each bit being independent of the others. This property can better preserve the original locality structure of the data but may render it unsuitable to calculate the distance between two hash codes as done in prior knowledge distillation methods, which are generally designed for real-value representations [46, 51, 61].

To realize the above factors while addressing the related challenges, we propose a novel Bit-mask Robust Contrastive knowledge Distillation (BRCD) method. **First**, we achieve the semantic space alignment for unsupervised semantic hashing by identifying two learning targets, including individual-space knowledge distillation and structural-semantic knowledge distillation. Specifically, the former forces the student model to learn embedding position knowledge from the teacher model for the same image. The latter ensures the student model acquires structure relation knowledge between different images from the teacher model. We introduce a contrastive knowledge distillation method to achieve both targets and demonstrate its effectiveness through formal analysis. **Second**, in our contrastive knowledge distillation, we try to generate positive samples through data augmentation. However, the teacher model may assign some augmented images to a distant location from the anchor images as illustrated in Figure 2 (a), where image $A$ and its augmentation $A'$ may not be close in Hamming space. We name such samples as "offset positive samples". This issue arises because augmented images may represent out-of-distribution data

for the teacher model. Served as noisy data, these offset positive samples provide wrong optimization directions. Thus, a cluster-based method is employed in the contrastive knowledge distillation process to detect and remove offset positive samples explicitly. The new contrastive method protects the student model from being influenced by the wrong optimization direction and improves the robustness of the knowledge distillation. **Third**, we conduct a quantitative analysis of bit independence at the bit level and reveal an interesting problem of "redundancy bit" presence in hash codes. This issue indicates that some bits in hash codes are redundant for a specific relevance set under the assumption of bit independence, leading to decreased learning effectiveness. Therefore, we present a bit mask mechanism to revise similarity calculations to mitigate the effects of redundancy bit. Finally, extensive experiments demonstrate our BRCD can achieve significant improvements over several knowledge distillation baselines and also show the generality of BRCD across diverse semantic hashing models and backbones.

## 2 RELATED WORK

### 2.1 Hashing-based Image Retrieval

Recently, hashing has increasingly gained importance in large-scale image retrieval. They can be broadly classified into two categories: supervised [2, 25, 44, 47, 57] and unsupervised [17, 18, 22, 27, 28, 52, 58]. The primary distinction between these two approaches is the availability of supervised information. In this paper, we focus on unsupervised hashing methods, as they effectively leverage unlabeled data and enable practical applications.

Unsupervised hashing methods try to preserve similarities of original data in the Hamming space. With the development of deep learning, deep semantic hashing has attracted growing interest in efficient image search. These methods can be roughly classified into two categories. Firstly, weak supervised learning-based approaches aim to reconstruct similar structures using pre-trained models [17, 38, 58, 59] or cluster methods [27, 28, 52]. These reconstructed structures or labels guide hash code learning via deep supervised hashing methods. Some of them [19, 52, 53] incorporate the idea of knowledge distillation, but their objective is to construct pseudo labels or signals rather than focusing on reducing the inference time. Secondly, self-supervised learning-based techniques incorporate popular self-supervised methods such as auto-encoders [5, 39] and generative adversarial networks (GANs) [6, 42] into deep unsupervised hashing. Recently, several methods have adopted contrastive learning in unsupervised hashing [18, 28, 34, 49]. For example, CIBHash [34] applies contrastive learning to unsupervised hashing from an information bottleneck perspective.

With the development of large backbones like ViT [8], some advanced works [10, 19] report a great improvement in semantic hashing when using such a powerful backbones. Unfortunately, the computational overhead of large-scale backbones conflicts with the efficiency requirements of semantic hashing. Therefore, there is a need to discover ways to reduce inference time in practice.

### 2.2 Knowledge Distillation in Image Retrieval

The concept of knowledge distillation, which involves transferring knowledge from one model (teacher) to another (student), has

gained widespread attention in recent years [11]. The primary challenge is to determine what knowledge the teacher has learned and how to distill it to the student [16]. This paper focuses on exploring knowledge distillation techniques suitable for image retrieval tasks.

Analyzing and exploiting the relation between data samples is a popular approach in image retrieval. In knowledge distillation, similarity-preserving knowledge (SP) [48] and relational knowledge distillation (RKD) [32] aim to transfer knowledge by preserving the sample relations between input pairs. However, they use shallow relation modeling between features, which may be suboptimal for capturing complex inter-sample relations. To address this issue, some studies have integrated contrastive learning into knowledge distillation [46, 51, 55, 61]. For example, CRD [46] combines knowledge distillation with contrastive learning to maximize the mutual information between teacher and student representations. Additionally, SSKD [51] employs contrastive tasks as self-supervised pretext tasks to enable richer knowledge extraction from the teacher to the student. CRCD [61] also utilizes contrastive loss but focuses on the mutual relations of deep representations instead of the representations themselves. PACKD [55] further considers the correlation among intra-class samples. These approaches enable more effective modeling of complex relations for improved knowledge distillation in image retrieval.

Our proposed knowledge distillation approach differs from existing methods in several aspects. First, it is reasonable to consider transferring the relation between data samples in the symmetric paradigm (SSHP). However, in the asymmetric paradigm (ASHP), the student model is exclusively utilized in the online stage, while the teacher model is used in the offline stage. As such, solely transferring relations may lead to invalid search results because of the inconsistent output spaces between the two models. To overcome this limitation, we develop a novel contrastive knowledge distillation method that accounts for the unique characteristics of the ASHP. Besides, existing knowledge distillation methods are designed for real-value representation, which does not consider some special problems in semantic hashing or property in hash codes. We modify our knowledge distillation method to cope with these special challenges in the semantic hashing domain.

## 3 PRELIMINARIES

In this section, we will briefly introduce semantic hashing and the semantic hashing search paradigms SSHP and ASHP.

### 3.1 Semantic Hashing

Consider a database $X = \{x_1, ..., x_N\}$ comprising $N$ images. Semantic hashing targets to learn a hash function $f : x \mapsto h$ that maps each image $x$ to a low-dimensional binary vector $h \in \{-1, 1\}^b$, referred to as a hash code, where $b$ denotes the dimensionality of $h$. This mapping aims to preserve the pairwise similarities between the images $x_i$ and $x_j$ in the Hamming space, characterized by the Hamming distance $D_H(h_i, h_j)$ for hash codes $h_i$ and $h_j$.

Several methods [22, 34, 39, 43, 49] can be used to generate discrete outputs. For example, a popular method is incorporating soft activation like $tanh(\cdot)$ to generate the output $v = g(x) \in (-1, 1)^b$ first. In the test process, input $v$ to the $sign(\cdot)$ function and get the final binary codes $h = sign(v)$. Another conventional method is

incorporating a probabilistic binary representation layer into the model. This involves first computing the output $v = g(x) \in (0, 1)^b$ using the feature extractor backbone. Next, we treat $v$ as the probability of a multivariate Bernoulli distribution, from which we can sample each bit $h_i \sim \text{Bernoulli}(v_i)$. To estimate the gradient of neural networks containing discrete stochastic variables, one common approach is to use the Straight-Through estimator [54]. In addition to these two primary methods, there are also several alternative approaches [22, 39, 43]. Our distillation technique holds potential for application across a broad spectrum of semantic hashing models, as will be demonstrated in Section 5.5.

### 3.2 Semantic Hashing Search Paradigm with Knowledge Distillation

When applying knowledge distillation methods to semantic hashing, we identify two search paradigms, including the Symmetric Semantic Hashing search Paradigm (SSHP) and the Asymmetric Semantic Hashing search Paradigm (ASHP).

The search paradigm typically involves online and offline stages. When using the KD technology, we have a pre-trained large teacher model $f_t :\mapsto h^t$ and a lightweight student model $f_s : x \mapsto h^s$ guided by the teacher model $f_t$. In the SSHP, the student model $f_s$ is used to precompute hash codes for candidate images, which can be used to construct the semantic hashing index [31]. In the online stage, the student model $f_s$ is also used to extract the hash codes of query images, and the index is utilized to locate relevant images quickly. ASHP introduces a modification to the SSHP. ASHP still employs a lightweight student model $f_s$ during the online stage but uses the large teacher model $f_t$ during the offline stage. Due to this setting, ASHP can rapidly return results in the online stage while obtaining more accurate hash codes in the offline stage.

However, to make the SSHP and ASHP effective for semantic hashing tasks, we must design an appropriate knowledge distillation method that achieves semantic space alignment and solves critical issues in the distillation process and hash codes. We describe our method in the following section.

## 4 THE PROPOSED BRCD METHOD

In this section, we present a description of our proposed Bit-mask Robust Contrastive knowledge Distillation (BRCD).

### 4.1 Contrastive Knowledge Distillation

To achieve valid results in both SSHP and ASHP, the basic objective of knowledge distillation is semantic space alignment, which consists of two targets. Firstly, the student model should learn the individual-space knowledge from the teacher model $f_t$. It targets to force hash codes of the same image $x_i$ output in student model $f_s$ and teacher model $f_t$ close to each other. This can be formulated as follows:

$$f_s = argmin_{f_s} \sum_{x_i \in X} D_H(f_s(x_i), f_t(x_i)), \quad (1)$$

where $D_H(\cdot, \cdot)$ represents a Hamming distance measurement, and the lower the value, the more similar the compared images. Secondly, the student model should learn the structural-semantic knowledge from the teacher model. The goal is to ensure that image pairs

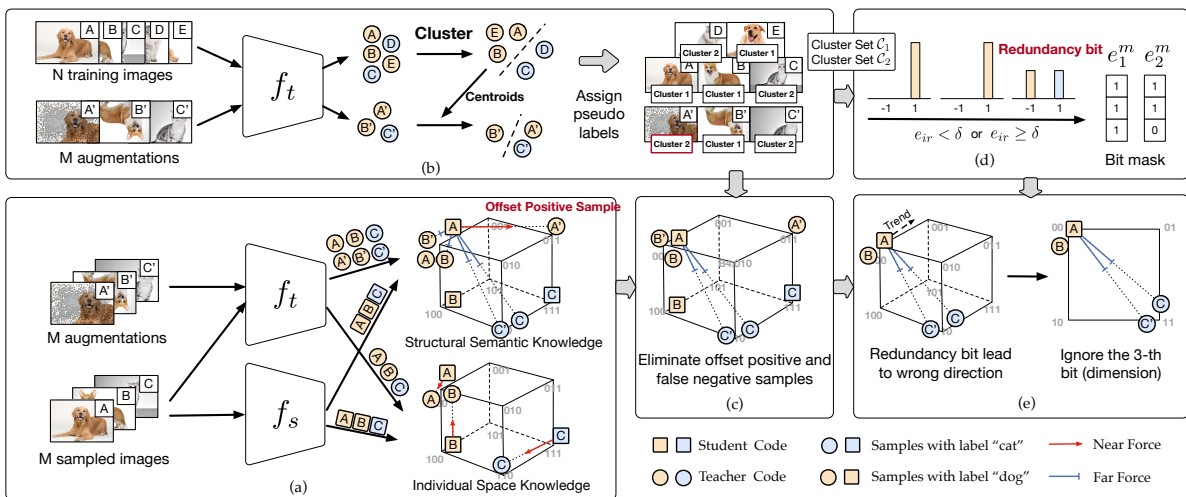

**Figure 2: Workflow of BRCD. (a) The contrastive knowledge distillation achieves individual-space and structural-semantic knowledge transfer. (b) Clustering and assigning pseudo labels to images. (c) Cluster-based method eliminates offset positive and false negative samples. (d) The process to get bit masks. (e) Bit mask mechanism prevents incorrect optimization direction.**

$(x_i, x_p)$ with similar semantics are mapped closely in the Hamming space while pairs $(x_i, x_n)$ with dissimilar semantics are far apart when they are inputted into student model $f_s$ and teacher model $f_t$. This objective can be expressed as follows:

$$f_s = argmin_{f_s} \sum_{x_i \in X} \left( \sum_{x_p \in P(i)} D_H(f_s(x_i), f_t(x_p)) \right.$$
$$\left. - \sum_{x_n \in N(i)} D_H(f_s(x_i), f_t(x_n))) \right), \quad (2)$$

where $N(i)$ denotes the negative set for $x_i$, $P(i)$ represents the positive set for $x_i$. Considering either of the two objects separately is not sufficient. If we only consider individual-space knowledge, it is hard to optimize all $x_i \in X$ to achieve the Eq. (1) because of the capacity gap between student and teacher models [30]. Conversely, concentrating solely on structural-semantic knowledge, the original space position is ignored and may lead to suboptimal results.

To design a knowledge distillation objective to satisfy both Eq. (1) and Eq. (2), we still lack information on $N(i)$ and $P(i)$ in the unsupervised scenario. Inspired by SimCLR [3], we utilize the augmented images as the positive images of the anchor while including other images from a given batch as irrelevant images. As illustrated in Figure 2 (a), given $M$ randomly sampled images from the training set $X$, our contrastive training mini-batch consists of $2M$ images obtained by applying data augmentation on the sampled image. Then we get original sample set $B = \{x_1, ..., x_M\}$ and augmented sample set $B' = \{x_{1'}, ..., x_{M'}\}$. The only relevant (positive) image for $x_i$ is its augmentation, denoted as $x_{i'}$, while the other $2(M-1)$ images jointly form the set of negative samples $N(i)$. Using the teacher model, we obtain sample hash code set $H_t = \{h_1^t, ..., h_M^t\}$ and augmented sample hash code set $H_t' = \{h_{1'}^t, ..., h_{M'}^t\}$. By inputting training batch $B$ into the student model $f_s$, we get $H_s = \{h_1^s, ..., h_M^s\}$. Subsequently, we propose a contrastive loss function as follows:

$$L = \sum_{i=1,..,M} -log \frac{exp((\alpha \cdot \phi(h_i^s, h_i^t) + (1-\alpha) \cdot \phi(h_i^s, h_{i'}^t))/\tau)}{\sum_{r \in R(i)} exp(\phi(h_i^s, h_r^t)/\tau)}. \quad (3)$$

Here, $\phi(\cdot, \cdot)$ is the cosine similarity, $R(i) = \{N(i), i, i'\}$ and $\alpha \in [0, 1]$ is a hyper-parameter that controls whether the hash code $h_i^s$ should be closer to $h_i^t$ or $h_{i'}^t$. To explore its behaviors, we derive the gradients with respect to the hash code $h_i^s$ as:

$$\frac{\partial L_i}{\partial h_i^s} = \sum_{n \in N(i)} \frac{\rho_1}{\tau} \cdot h_n^t - \frac{\alpha \rho_2}{\tau} \cdot h_i^t - \frac{(1-\alpha)\rho_3}{\tau} \cdot h_{i'}^t, \quad (4)$$

where $\rho_1, \rho_2, \rho_3$ are the weighted coefficients. Based on their gradients in Eq. (4), we can find the object of Eq. (3) is the generalization of the combination between the knowledge distillation targets described in Eq. (1) and Eq. (2) with coefficients (detailed proof is presented in Appendix A). Thus, it can facilitate the student model's learning of individual-space knowledge and structural-semantic knowledge from the teacher model in Hamming space as shown in Figure 2 (a).

## 4.2 Cluster-based Robust Optimization

Although we have proposed the contrastive knowledge distillation Eq. (3) to support the SSHP and ASHP, relying solely on the proposed contrastive loss can not ensure a robust knowledge distillation process. For example, the hash code of anchor image $x_i$ and its augmentation $x_{i'}$ should ideally be close in Hamming space. However, as depicted in the upper-right part of Figure 2 (a), due to the teacher model may produce inaccurate hash codes for augmentations, the hash codes of image $A$ and its augmentation $A'$ may not be close in Hamming space. Consequently, these augmentations serve as noisy data, and the optimization may force the student model in the wrong direction according to Eq. (3). We term these augmentations as "offset positive samples" and evaluate their occurrence probability in Appendix D.

To ensure a robust optimization process, we propose a cluster-based method that explicitly detects and removes offset positive samples to learn the semantic output space of the teacher model effectively. As illustrated in Figure 2 (b), we first perform k-means clustering on the set of hash codes $H_t^{all} = \{h_1^t, ..., h_N^t\}$ of all training

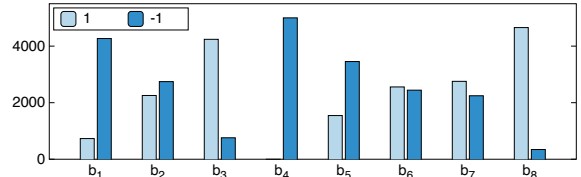

**Figure 3: We conduct the analysis using 5000 images with the same class on the CIFAR-10 dataset. It shows the frequency histograms of eight randomly chosen dimensions.**

images and group them into $k$ clusters. In the unsupervised scenario, we can determine the value of $k$ by using the elbow method or silhouette analysis [35]. Subsequently, we assign a pseudo label $y_i$ to image $x_i$ based on the closest centroid. In each training batch, augmentations $B' = \{x_{1'}, ..., x_{M'}\}$ are also assigned their respective pseudo label $y_{i'}$ based on the closest centroid. If the anchor image $x_i$ and its augmentation $x_{i'}$ have different pseudo labels, we consider $x_{i'}$ as an offset positive sample. Thus, we define a dynamic $\alpha'_i$ to replace the original $\alpha$ in Eq. (3) as follows:

$$\alpha'_i = \begin{cases} \alpha, & y_i = y_{i'} \\ 1, & y_i \neq y_{i'}. \end{cases} \tag{5}$$

Additionally, this method can incidentally solve the problem of false negative samples [37]. The negative sample set $N(i)$ is random and includes semantic content similar to the anchor image $x_i$. This inclusion can cause a significant distance between one image and its relevant image, thereby adversely affecting the optimization process. If two different images $x_i$ and $x_j$ belong to the same centroid in one training batch, we consider $x_j$ a false negative sample of $x_i$. Thus, we modify the set $R$ in Eq. (3) as follows:

$$R'(i) = \{k, i, i' | k \in N(i), y_k \neq y_i\}. \tag{6}$$

Finally, the contrastive loss becomes:

$$L = \sum_{i=1,..,M} -\log \frac{exp((\alpha'_i \phi(h_i^s, h_i^t) + (1 - \alpha'_i)\phi(h_i^s, h_{i'}^t))/\tau)}{\sum_{r \in R'(i)} exp(\phi(h_i^s, h_r^t)/\tau)}. \tag{7}$$

The new contrastive loss is effective in detecting and removing the offset positive sample and false negative sample issues depicted in Figure 2 (c), where image $A$ eliminates the impact of offset positive sample $A'$ and does not consider its relation with false negative samples $B$ and $B'$. Thus Eq. (7) can alleviate the influence of noisy samples and achieve a robust knowledge distillation process.

### 4.3 Bit mask mechanism

To achieve semantic space alignment, we need to measure the similarity between two hash codes, where we apply cosine similarity $\phi(\cdot, \cdot)$ in the learning objective Eq. (7). However, current approaches may overlook the special distribution property of hash codes referred to as "bit independence." Specifically, bit independence means each bit is independent of the others in the output distribution for all hash codes [13]. It can better preserve the original locality structure of the data [7] but may render it unsuitable to calculate the similarity as prior knowledge distillation methods directly. To address this property, we first explore the distribution of bits within one relevance set.

Specifically, we obtain $H_t^{all} = \{h_1^t, ..., h_N^t\}$ by using the candidate set $X$ and the trained teacher model $f_t$. We then generate relevance sets $\{G_1, ..., G_k\}$, $G_m = \{h_1^m, ..., h_{|G_m|}^m\}$, $h_i^m = [h_{i1}^m, ..., h_{ib}^m]$ based on their ground truth (e.g., class), where $k$ is the number of relevance sets, $|G_m|$ is the number of elements in $G_m$ and $h_{ij}^m \in \{-1, 1\}$ is the j-th bit of hash code $h_i^m$. We randomly choose a relevance set $G_m$ and compute the frequency of 1 and -1 in each dimension for all hash codes in $G_m$ to plot the frequency histograms. Figure 3 shows the frequency histogram of eight randomly selected dimensions. We can observe two types of bits: (1) in some dimensions, the frequency of 1 and -1 is disparate (e.g., $b_1$, $b_3$, $b_4$, $b_5$, $b_8$), and (2) in other dimensions, the frequency of 1 and -1 is slightly disparate (e.g., $b_2$, $b_6$, $b_7$). Under the bit independence assumption, if 1 and -1 have an equal probability of occurring in a specific dimension within a relevance set, this dimension does not provide any useful semantic information because it tends to express random semantics. Worse still, these dimensions lead to the wrong optimization direction for structural knowledge in the learning process. As shown in the left part of Figure 2 (e), to stay away from images $C$ and $C'$, image $A$ tends to move away from its relevant images $B$. We refer to such bit as "**redundancy bit**".

To mitigate the bad effect of redundancy bits, we propose a bit mask mechanism that excludes redundancy bits in similarity calculation, as shown in Figure 2 (d). In the unsupervised scenario, we use the cluster results of all training hash codes $H_t^{all}$ obtained in Section 4.2 to get $k$ cluster sets $\{C_1, ..., C_k\}$ as relevance sets. Here, $C_i = \{h_1^i, ..., h_{|C_i|}^i\}$ and $|C_i|$ is the number of elements in $C_i$. Next, we calculate the expectation $e_{ir}$ of each bit for each relevance set $C_i$ as follows:

$$e_{ir} = \frac{1}{|C_i|} |\sum_{j=1}^{|C_i|} h_{jr}^i|, \tag{8}$$

where $h_{jr}^i$ is the r-th dimension of $h_j^i$. Then, we can define the i-th cluster's bit mask as $e_i^m = [e_{i1}^m, ..., e_{ib}^m]$, where the r-th bit mask $e_{ir}^m$ for relevance set $C_i$ is defined as:

$$e_{ir}^m = \begin{cases} 1, & e_{ir} \geq \delta \\ 0, & e_{ir} < \delta. \end{cases} \tag{9}$$

Here, $\delta$ is a threshold. Then, we propose the Bit-mask Robust Contrastive knowledge Distillation (BRCD) as follows:

$$L = \sum_{i=1,...,M} -\log \frac{exp((\alpha'_i \phi(h_i^s, h_i^t) + (1 - \alpha'_i)\varphi(h_i^s, h_{i'}^t))/\tau)}{exp(\phi(h_i^s, h_i^t)) + \sum_{r \in \hat{R}(i)} exp(\varphi(h_i^s, h_r^t)/\tau)}, \tag{10}$$

where $\hat{R}(i) = \{k, i' | k \in N(i), y_k \neq y_i\}$ and $\varphi(\cdot, \cdot)$ represents the revised similarity function, which is defined as:

$$\varphi(a, b) = \phi(e_{c(a)}^m \cdot a, e_{c(b)}^m \cdot b). \tag{11}$$

Here, $e_{c(a)}^m$ is the bit mask of the relevance set corresponding to image $a$. The motivation behind it is that if the expectation of one dimension in a relevance set is lower than the threshold $\delta$, it should be the redundancy bit. In Eq. (10), when measuring structural-semantic knowledge, the dimensions of redundancy bits are ignored. This setting prevents the optimization process from the suboptimal results. For instance, in the right part of Figure 2 (e), image $A$ avoids choosing the wrong optimization direction after eliminating the redundancy bit. Certainly, the assumption of "each bit is independent

of the other" is a relative concept. Many semantic hashing models [22, 34, 60] approach this target, but the complete bit independence is hard to grasp. Therefore, the hyper-parameter $\delta$ in Eq. (9) can be considered a reflection of bit independence. If a semantic hashing model can achieve bit independence well, a high value should be given to $\delta$; otherwise, a small value is appropriate. We summarize the training algorithm of BRCD in Appendix C

## 5 EXPERIMENTS

### 5.1 Dataset

We conduct experiments on three datasets to evaluate the performance of our proposed hashing method. **CIFAR-10** [20] consists of 60,000 images from 10 classes. We randomly select 1,000 images per class as the query set, 500 images per class as the training set, and use all remaining images except queries as the database. **MSCOCO** [24] is a large-scale dataset for object detection, segmentation, and captioning. We consider a subset of 122,218 images from 80 categories, as in previous works [34]. We randomly select 5,000 images from the subset as the query set and use the remaining images as the database. For training, we randomly select 10,000 images from the database. **ImageNet100** is a subset of ImageNet with 100 classes. We follow the settings from [1, 9] and randomly select 100 categories. Then, we use all the images of these categories in the training set as the database and the images in the validation set as the queries. Furthermore, we randomly select 13,000 as the training images from the database.

### 5.2 Evaluation Metric

We use the mean Average Precision (mAP) at the top K as the evaluation metric in our experiments. Specifically, we adopt mAP@5000 for MSCOCO, mAP@1000 for CIFAR-10, and ImageNet100, following the settings used in previous work [9, 34].

### 5.3 Training Details

We implement models using Pytorch and conduct experiments on two Intel Xeon Gold 5218 CPUs and one NVIDIA Tesla V100. Model training consists of two parts: the training loss in the semantic hashing model itself and the knowledge distillation loss. In our knowledge distillation part, we implement the optimizer Adam for optimization, in which the default parameters are used, and the learning rate is set to 0.001. The temperature $\tau$ is set to 0.3, and we consider the hyper-parameters $\alpha$ in $\{0.6, 0.7, 0.8, 0.9\}$ and $\delta$ in $\{0.2, 0.3, 0.4, 0.5, 0.6\}$. We perform the grid search method on different cases for the best combination.

### 5.4 Knowledge Distillation Methods Comparison

In this experiment, we compare the mAP@K of different knowledge distillation methods on three datasets.

*5.4.1 Setting.* We consider the following knowledge distillation methods for comparison: KL [16], SP [48], RKD [32], PKT [33], CRD [46], SSKD [51], CRCD [61] and PACKD [3] [55]. These methods, including BRCD, are applied to the final output of the semantic

---

[3]PACKD requires the availability of supervised signals to construct positive data, and we have applied this setting in our experiments.

hashing model, i.e., the hash code layer for practicability. We choose CIBHash [34] as the semantic hashing model to train the teacher and student models. We use ViT_B_16 [8] as the teacher model's backbone and use EfficientNetB0 [45] as the student model's backbone. The code length $b$ is set to 32 and 64. To explore the effect of different components in the BRCD method, we design variants of BRCD, including BRCD w/o NP (BRCD without removing false negative and offset positive samples), BRCD w/o BM (BRCD without bit mask mechanism) and BRCD w/o P (BRCD without removing offset positive samples).

*5.4.2 Results.* Table 1 summarizes the results, where we make the following observations:

1) Our proposed BRCD method outperforms other baselines. In the SSHP, BRCD performs competitively with other baselines. These results demonstrate that our BRCD method effectively captures and transfers valuable knowledge to enhance search performance. Furthermore, in the ASHP, BRCD shows significant performance, surpassing the second-best result by 9.6%, 4.3%, and 3.1% on the CIFAR-10, ImageNet100, and MSCOCO datasets, respectively, averaged across two code length. We attribute this success to the specific considerations made by BRCD, such as semantic space alignment between the student and teacher models, a robust optimization procedure, and the incorporation of hash code properties.

2) Certain baselines exhibit better performance in the ASHP compared to the SSHP, while others fail to reach the same level of effectiveness. For instance, KL is effective in the ASHP because it promotes semantic space alignment by reducing the distributional discrepancies between binary representations. Conversely, methods like RKD, PKT, CRD, SSKD, CRCD, and PACKD do not achieve semantic space alignment as they primarily focus on transferring relations between images, resulting in their inability to yield valid results in the ASHP. Interestingly, in ASHP, SP performs well in some situations but fails in others. Through formal analysis presented in Appendix B, we have discovered that the optimization of the SP method exhibits two directions in the Hamming space: one direction aims to achieve individual-space alignment, while the other does not. This observation explains the divergent outcomes achieved by the SP method in the ASHP.

3) Compared to BRCD, BRCD w/o NP, BRCD w/o P, and BRCD w/o BM have poor performance, indicating that the consideration of false negative and offset positive samples, as well as the bit mask mechanism, is necessary to improve the performance for our knowledge distillation methods. Besides, compared to BRCD w/o NP, BRCD w/o P achieves significant improvement. These results confirm the importance of considering offset positive samples for effective knowledge distillation. It is worth noting that the issue of offset positive samples has been largely overlooked in previous studies. In Appendix D, we conducted further analysis on offset positive samples to validate the prevalence of this phenomenon.

## 5.5 Performance on Different Hashing Models

In this experiment, we further validate BRCD on different hashing models by using mAP@1000 on the CIFAR-10 dataset.

**Table 1: The mAP@K comparison results on CIFAR-10, MSCOCO, and IMAGENET100 when using different knowledge distillation methods. The best result in each column is marked with bold. The second-best result in each column is underlined.**

| KD Methods | CIFAR-10 (mAP@1000) | | | | MSCOCO (mAP@5000) | | | | IMAGENET100 (mAP@1000) | | | |
|---|---|---|---|---|---|---|---|---|---|---|---|---|
| | 32 bit | | 64 bit | | 32 bit | | 64 bit | | 32 bit | | 64 bit | |
| | SSHP | ASHP | SSHP | ASHP | SSHP | ASHP | SSHP | ASHP | SSHP | ASHP | SSHP | ASHP |
| NO KD | 0.4935 | - | 0.5111 | - | 0.7459 | - | 0.7609 | - | 0.7097 | - | 0.7613 | - |
| KL [16] | 0.6300 | 0.6612 | 0.6460 | 0.6767 | 0.7956 | 0.8013 | 0.8087 | 0.8120 | 0.8463 | 0.8568 | 0.8576 | 0.8671 |
| SP [48] | 0.6208 | 0.4728 | 0.6273 | 0.6765 | 0.7984 | 0.8123 | 0.8129 | 0.7320 | 0.8524 | 0.7473 | 0.8641 | 0.4031 |
| PKT [33] | 0.5939 | 0.1075 | 0.6035 | 0.1101 | 0.7532 | 0.3499 | 0.7602 | 0.3376 | 0.8241 | 0.0134 | 0.8365 | 0.0129 |
| RKD [32] | 0.4059 | 0.1089 | 0.6171 | 0.0969 | 0.7813 | 0.3785 | 0.7951 | 0.3843 | 0.8319 | 0.0164 | 0.8496 | 0.0155 |
| CRD [46] | 0.6366 | 0.0982 | 0.6549 | 0.1117 | 0.7934 | 0.3395 | 0.8013 | 0.3356 | 0.8346 | 0.0126 | 0.8479 | 0.0145 |
| SSDK [51] | 0.6086 | 0.0986 | 0.6268 | 0.1457 | 0.7846 | 0.3596 | 0.7967 | 0.3583 | 0.8158 | 0.0156 | 0.8328 | 0.0164 |
| CRCD [61] | 0.6382 | 0.0922 | 0.6576 | 0.0927 | 0.7986 | 0.3256 | 0.8059 | 0.3285 | 0.8394 | 0.0124 | 0.8589 | 0.0126 |
| PACKD [55] | 0.6453 | 0.1009 | 0.6699 | 0.0657 | 0.8089 | 0.0343 | 0.8204 | 0.0341 | 0.8574 | 0.0138 | 0.8693 | 0.0129 |
| BRCD | **0.6787** | **0.7285** | **0.6900** | **0.7378** | **0.8208** | **0.8336** | **0.8349** | **0.8407** | **0.8866** | **0.8955** | **0.8966** | **0.9020** |
| BRCD w/o BM | 0.6612 | 0.7100 | 0.6721 | 0.7216 | 0.8087 | 0.8233 | 0.8251 | 0.8306 | 0.8761 | 0.8843 | 0.8860 | 0.8919 |
| BRCD w/o NP | 0.6564 | 0.7068 | 0.6659 | 0.7178 | 0.8074 | 0.8219 | 0.8239 | 0.8287 | 0.8715 | 0.8804 | 0.8818 | 0.8821 |
| BRCD w/o P | 0.6650 | 0.7144 | 0.6756 | 0.7259 | 0.8156 | 0.8253 | 0.8274 | 0.8329 | 0.8789 | 0.8872 | 0.8885 | 0.8945 |

**Table 2: The mAP@1000 comparison results on CIFAR-10 when using different hashing models and knowledge distillation methods. The best result in each column is marked with bold. The second-best result in each column is underlined.**

| Hashing Model | Paradigm | NO KD | KL | SP | PKT | RKD | CRD | SSDK | CRCD | PACKD | BRCD |
|---|---|---|---|---|---|---|---|---|---|---|---|
| GreedyHash [43] | SSHP | 0.3084 | 0.6128 | 0.6284 | 0.5784 | 0.6013 | 0.6295 | 0.6005 | 0.6453 | 0.6481 | **0.6687** |
| | ASHP | - | 0.6651 | 0.6525 | 0.1204 | 0.1123 | 0.1142 | 0.1347 | 0.1023 | 0.0924 | **0.7023** |
| Bi-half Net [22] | SSHP | 0.3254 | 0.5680 | 0.6414 | 0.5481 | 0.6171 | 0.6601 | 0.6125 | 0.6669 | **0.6721** | 0.6703 |
| | ASHP | - | 0.6189 | 0.4199 | 0.1252 | 0.0978 | 0.0940 | 0.1274 | 0.0936 | 0.1162 | **0.7085** |
| TBH [39] | SSHP | 0.4383 | 0.6215 | 0.6337 | 0.5794 | 0.6326 | 0.6403 | 0.6052 | 0.6475 | 0.6518 | **0.6755** |
| | ASHP | - | 0.6536 | 0.6635 | 0.1217 | 0.1126 | 0.1278 | 0.1336 | 0.0974 | 0.0837 | **0.7153** |
| CIBHash [34] | SSHP | 0.5111 | 0.6460 | 0.6273 | 0.6035 | 0.6171 | 0.6549 | 0.6268 | 0.6576 | 0.6699 | **0.6900** |
| | ASHP | - | 0.6767 | 0.6765 | 0.1101 | 0.0969 | 0.1117 | 0.1457 | 0.0927 | 0.0657 | **0.7378** |
| MeCoQ [49] | SSHP | 0.5483 | 0.6693 | 0.6536 | 0.6398 | 0.6534 | 0.6873 | 0.6431 | 0.6882 | 0.7045 | **0.7228** |
| | ASHP | - | 0.6912 | 0.6935 | 0.1085 | 0.1027 | 0.1075 | 0.1367 | 0.0988 | 0.0894 | **0.7504** |

*5.5.1 Setting.* We still use the previous knowledge distillation methods for comparison and consider the following representative semantic hashing models: GreedyHash [43], TBH [39], Bi-half Net [22], CIBHash [34], and MeCoQ [49]. These methods utilize distinct binarization techniques. GreedyHash applies a hash layer with a sign function and uses a greedy algorithm for fast discrete optimization. Bi-half Net adopts a bi-half layer to generate balanced and discrete hash codes. TBH utilizes element-wise discrete stochastic neuron activation [5]. CIBHash samples discrete hash codes from the Bernoulli distribution. MeCoQ uses a soft activation to get the binary-like representation in the training stage. We still use ViT_B_16 as the teacher model's backbone and EfficientNetB0 as the student model's backbone. The code length $b$ is set to be 64.

*5.5.2 Results.* Table 2 summarizes the results. We can find the following observations:

1) In most cases, our proposed BRCD method is better than other baselines when using different hashing models. In SSHP, the BRCD is superior to all KD methods except for PACKD when using the Bi-harf Net model. However, PACKD applies supervised signals to create positive data, whereas our proposed BRCD does not require any supervised signal at all. In ASHP, BRCD is better than the other

methods in these five hash models. This result demonstrates the generality of our method across different semantic hash models.

2) Despite our initial intention to use distillation for accelerating the inference process, the results indicate that employing distillation methods can effectively improve the performance of most semantic hashing models. We can observe that even when using the traditional KL method, there is a minimum improvement of 22% (this is also due to ViT serving as the teacher model backbone). The phenomenon highlights the importance of distillation techniques in unsupervised semantic hashing methods.

## 5.6 Performance on Different Backbones

In this experiment, we validate our methods when using different backbones on the CIFAR-10 dataset. Specifically, we adopt several representative knowledge distillation methods that have been used earlier for comparison. The teacher model uses ViT_B_16 [8] or ViT_L_16 as backbone, and the student model uses EfficientNetB0 [45], ResNet18 [14], or MobileNetV2 [36] as backbone. In prior experiments, our proposed BRCD achieves a significant advantage within the ASHP and most KD methods can't get valid performance in this paradigm. Consequently, we select some representative KD methods for comparison under the SSHP in this experiment. We

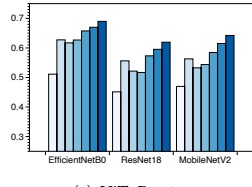
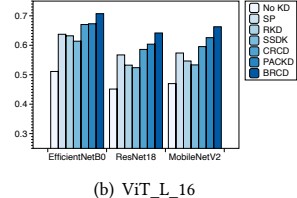

(a) ViT_B_16

(b) ViT_L_16

**Figure 4: The mAP@1000 results on the CIFAR-10 dataset when using different knowledge methods and backbones.**

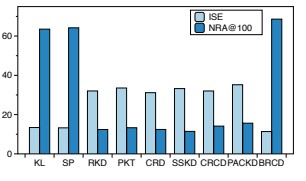
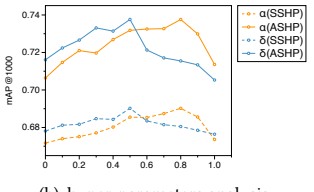

(a) semantic space alignment analysis

(b) hyper-parameters analysis

**Figure 5: (a) The ISD and NRA@K on CIFAR-10 dataset for semantic space alignment analysis. (b) Hyper-parameters analysis on $\alpha$ and $\delta$.**

perform this setting in the CIFAR-10 dataset, use CIBHash as the hashing model, and set the code length to 64. Figure 4 shows the performance, from which we can find that BRCD is still better than other KD methods when using different backbones in the teacher model or student model. This outcome validates the generality of our method across diverse backbone scenarios.

### 5.7 Model Analysis

*5.7.1 Analysis on semantic space alignment.* Semantic space alignment is a prerequisite in the ASHP and a crucial factor for SSHP. To verify the ability of the BRCD from the perspective of semantic space alignment, we analyze the distribution of hash codes from the metric calculation. We propose two metrics to evaluate semantic space alignment from two perspectives. Moreover, a visualization analysis is shown in Appendix E.

First, to assess the transfer of individual-space knowledge, we introduce the Individual Sample Distance (ISD) metric, which measures whether the same image is assigned to a nearby space from different models. The formula for ISD is as follows:

$$ISD = \frac{\sum_{i=1,2,...,N}(b - f_s(x_i) \cdot f_t(x_i))}{2N}, \qquad (12)$$

where the lower the value of $ISD$, the closer the student's hash code and the teacher's hash code for the same image. Second, to evaluate structural knowledge transfer, we define the Neighbor Relevance Accuracy (NRA@K) metric as follows:

$$NRA@K = \sum_{i=1,...,N} |\{x_j | x_j \in \mathcal{N}_K(f_s(x_i)), y_j = y_i\}|, \qquad (13)$$

where $\mathcal{N}_K(a)$ means choosing the $K$ nearest neighbor of hash code $a$ among the set $H_t^{all} = \{h_1^t, h_2^t, ..., h_N^t\}$ in Hamming space. This metric measures the percentage of relevant images' codes from the teacher model around the hash codes produced by the student model, with a higher value indicating better structural-semantic knowledge transfer. We set $K$ to 100, perform this experiment on the CIFAR-10 dataset, and use ViT_B_16 as the teacher model's backbone and EfficientNetB0 as the student's backbone.

Figure 5(a) shows the results. We can find that KL, SP, and BRCD methods achieve lower $ISD$ than other distillation methods, with the BRCD achieving the lowest value. These findings suggest that our distillation method effectively achieves individual space knowledge transfer. Besides, our proposed BRCD also achieves the highest $NRA@K$. Thus, we have confidence that our proposed distillation method can effectively transfer structural-semantic knowledge. Moreover, the KL method achieves high $NRA@K$ without a

structural-semantic knowledge distillation objective. The phenomenon shows that individual space knowledge distillation can also partially achieve the goal of structural-semantic knowledge distillation. However, as discussed in Section 4.1, due to the capacity gap between student and teacher models, it is challenging to optimize all samples to achieve the ideal results of individual-space knowledge distillation. Therefore, we think the goal of structural-semantic knowledge distillation can also be regarded as a regularization term that prevents suboptimal cases during the optimization process of individual-space knowledge distillation.

*5.7.2 Parameter analysis.* To examine the impact of the key hyper-parameters $\alpha$ and $\delta$ on performance, we evaluate the model under different $\alpha$ and $\delta$ values. The parameter $\alpha$ in the Eq. (10) (specifically, the Eq. (10) is influenced by $\alpha$ through Eq. (5)) can adjust the objective function to prioritize either individual-space knowledge or structural-semantic knowledge. As shown in Figure 5 (b), $\alpha$ plays an important role in obtaining optimal performance. Setting $\alpha$ to values that are too small or too large fails to yield the best performance in either case. This observation underscores the importance of simultaneously considering both individual-space knowledge transfer and structural-semantic knowledge transfer. Besides, the parameter $\delta$ in Eq. (9) serves as the threshold for the bit mask. As parameter $\delta$ increases, more dimensions of the binary vector are masked. Similar to parameter $\alpha$, the parameter $\delta$ should not be set excessively large or small. When $\delta$ is set to a smaller value, it becomes insufficient to eliminate most of the redundancy bits. Conversely, if parameter $\delta$ is too large, it risks removing non-redundancy bits, resulting in the loss of valuable information.

## 6 CONCLUSION

In this paper, we investigated the practical problem of inference delay in semantic hashing and proposed a novel distillation method Bit-mask Robust Contrastive knowledge Distillation (BRCD). Our method introduces a contrastive knowledge distillation to ensure the effectiveness of the symmetric paradigm (SSHP) and the asymmetric paradigm (ASHP) in semantic hashing. Our method also provides a robust knowledge distillation process and eliminates the effect of redundancy bits. Extensive experiments on three datasets demonstrated the effectiveness of the BRCD methods. More importantly, we analyzed and discovered the "redundancy bit" property that exists in hash codes. Further studies on this property can be explored in the future.

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

## A THEORETICAL ANALYSIS FOR CONTRASTIVE KNOWLEDGE DISTILLATION LOSS

In Section 4.1, we have proposed a contrastive knowledge distillation objective in Eq.(3). In this section, we show the derivation of the gradients of this contrastive knowledge distillation objective and demonstrate it is the generalization of our two knowledge distillation targets. Without loss of generality, we use $i$ to represent an arbitrary anchor image and rewrite the loss function Eq.(3) as:

$$L_i = -log \frac{exp((\alpha h_i^s \cdot h_i^t + (1-\alpha)(h_i^s \cdot h_{i'}^t))/\tau)}{\sum_{r \in R(i)} exp(h_i^s \cdot h_r^t/\tau)}, R(i) = \{N(i), i, i'\}. \tag{14}$$

Next, we derive the gradient of $L_i$ with respect to the student hash codes of the anchor image $h_i^s$:

$$\begin{aligned}
\frac{\partial L_i}{\partial h_i^s} &= \frac{\partial}{\partial h_i^s} - log \frac{exp((\alpha h_i^s \cdot h_i^t + (1-\alpha)(h_i^s \cdot h_{i'}^t))/\tau)}{\sum_{r \in R(i)} exp(h_i^s \cdot h_r^t/\tau)} \\
&= \frac{\partial}{\partial h_i^s} log(\sum_{r \in R(i)} exp(h_i^s \cdot h_r^t/\tau)) \\
&\quad - \frac{\partial}{\partial h_i^s}(\alpha h_i^s \cdot h_i^t + (1-\alpha)(h_i^s \cdot h_{i'}^t))/\tau \\
&= \sum_{r \in R(i)} \frac{exp(h_i^s \cdot h_r^t/\tau)}{\sum_{r \in R(i)} exp(h_i^s \cdot h_r^t/\tau)} \cdot \frac{h_r^t}{\tau} - \alpha \cdot \frac{h_i^t}{\tau} - (1-\alpha) \cdot \frac{h_{i'}^t}{\tau} \\
&= \sum_{n \in N(i)} \frac{exp(h_i^s \cdot h_n^t/\tau)}{\sum_{r \in R(i)} exp(h_i^s \cdot h_r^t/\tau)} \cdot \frac{h_n^t}{\tau} \\
&\quad - (\alpha - \frac{exp(h_i^s \cdot h_i^t/\tau)}{\sum_{r \in R(i)} exp(h_i^s \cdot h_r^t/\tau)}) \cdot \frac{h_i^t}{\tau} \\
&\quad - ((1-\alpha) - \frac{exp(h_i^s \cdot h_{i'}^t/\tau)}{\sum_{r \in R(i)} exp(h_i^s \cdot h_s^t/\tau)}) \cdot \frac{h_{i'}^t}{\tau} \\
&= \sum_{n \in N(i)} \frac{\rho_1}{\tau} \cdot h_n^t - \frac{\alpha \rho_2}{\tau} \cdot h_i^t - \frac{(1-\alpha)\rho_3}{\tau} \cdot h_{i'}^t,
\end{aligned} \tag{15}$$

where $\rho_1$, $\rho_2$, and $\rho_3$ represent the coefficient terms defined as:

$$\rho_1 = \frac{exp(h_i^s \cdot h_n^t/\tau)}{\sum_{r \in R(i)} exp(h_i^s \cdot h_r^t/\tau)}, \tag{16}$$

$$\rho_2 = 1 - \frac{exp(h_i^s \cdot h_i^t/\tau)}{\alpha \sum_{r \in R(i)} exp(h_i^s \cdot h_r^t/\tau)}, \tag{17}$$

$$\rho_3 = 1 - \frac{exp(h_i^s \cdot h_{i'}^t/\tau)}{(1-\alpha) \sum_{r \in R(i)} exp(h_i^s \cdot h_s^t/\tau)}. \tag{18}$$

Please notice that the Hamming distance between two hash codes $h_a \in \{-1, 1\}^b$ and $h_b \in \{-1, 1\}^b$ can be described as:

$$D_H(h_a, h_b) = \frac{(b - h_a \cdot h_b)}{2}. \tag{19}$$

We then apply $\alpha$ and $(1 - \alpha)$ weights to individual-space knowledge and structural-semantic knowledge in Eqs. (1) and (2) of the manuscript, respectively. Consequently, we obtain the following equation:

$$\begin{aligned}
H_i &= \frac{\alpha}{2}(b - h_i^s \cdot h_i^t) + \frac{(1-\alpha)}{2} \sum_{p \in P(i)} (b - h_i^s \cdot h_p^t) \\
&\quad - \frac{1}{2} \sum_{n \in N(i)} (b - h_i^s \cdot h_n^t).
\end{aligned} \tag{20}$$

Similarly, the gradient of $H_i$ with respect to $h_i^s$ is given by:

$$\frac{\partial H_i}{\partial h_i^s} = \sum_{n \in N(i)} \frac{1}{2} h_n^t - \frac{\alpha}{2} \cdot h_i^t - \sum_{p \in P(i)} \frac{(1-\alpha)}{2} \cdot h_p^t. \tag{21}$$

It is worth noting that $P(i) = \{i'\}$ in our data augmentation setting. Thus, we can rewrite Eq.(21) as:

$$\frac{\partial H_i}{\partial h_i^s} = \sum_{n \in N(i)} \frac{1}{2} h_n^t - \frac{\alpha}{2} \cdot h_i^t - \frac{(1-\alpha)}{2} \cdot h_{i'}^t. \tag{22}$$

We can find Eq. (15) generalizes Eq. (22). Therefore, our objective functions possess implicit capabilities to facilitate the student model's learning of individual-space knowledge and structural-semantic knowledge from the teacher model in the Hamming space.

## B THEORETICAL ANALYSIS FOR SIMILARITY PRESERVE KNOWLEDGE DISTILLATION

In Section 5.4, we have found that in the asymmetric paradigm (ASHP), the results of the Similarity-Preserving knowledge distillation (SP) [48] method on different cases are vastly different. In this section, we present a formal analysis to explore why this phenomenon occurs. We represent the output of the student model and teacher model as binary hash codes $H_s = \{h_1^s, h_2^s, ..., h_n^s\} \in \{-1, 1\}^{n \times b}$ and $H_t = \{h_1^t, h_2^t, ..., h_n^t\} \in \{-1, 1\}^{n \times b}$, respectively, where $b$ denotes the length of the hash code. The objective of the SP method is defined as:

$$L_{sp} = \frac{1}{b^2} ||H_s H_s^T - H_t H_t^T||_F^2, \tag{23}$$

where $||\cdot||_F$ denotes the Frobenius norm. To simplify the analysis, we consider the relationship between hash codes of arbitrary two images' hash codes $h_i$ and $h_j$ and express the corresponding loss function as:

$$L_{sp}^{ij} = (h_i^s h_j^s - h_i^t h_j^t)^2. \tag{24}$$

We assume that $h_i = [h_{i1}, h_{i2}, ..., h_{ib}]$, where $h_{ik}$ represents the k-th bit of $h_i$. Expanding Eq. (24), we obtain:

$$\begin{aligned}
L_{sp}^{ij} &= (h_i^s h_j^s)^2 - 2(h_i^s h_j^s)(h_i^t h_j^t) + (h_i^t h_j^t)^2 \\
&= (h_{i1}^s h_{j1}^s + ... + h_{ib}^s h_{jb}^s)^2 + (h_{i1}^t h_{j1}^t + ... + h_{ib}^t h_{jb}^t)^2 \\
&\quad - 2(h_{i1}^s h_{j1}^s + ... + h_{ib}^s h_{jb}^s)(h_{i1}^t h_{j1}^t + ... + h_{ib}^t h_{jb}^t) \\
&= (h_{i1}^s h_{j1}^s)^2 + ... + (h_{ib}^s h_{jb}^s)^2 + (h_{i1}^t h_{j1}^t)^2 + ... + (h_{ib}^t h_{jb}^t)^2 \\
&\quad - 2(h_{i1}^s h_{j1}^s h_{i1}^t h_{j1}^t + ... + h_{ib}^s h_{jb}^s h_{ib}^t h_{jb}^t) \\
&\quad + 2 \sum_{k \neq r; k, r = 1, ..., b} (h_{ik}^s h_{jk}^s h_{ir}^s h_{jr}^s - h_{ik}^s h_{jk}^s h_{ir}^t h_{jr}^t \\
&\quad - h_{ik}^t h_{jk}^t h_{ir}^s h_{jr}^s + h_{ik}^t h_{jk}^t h_{ir}^t h_{jr}^t).
\end{aligned} \tag{25}$$

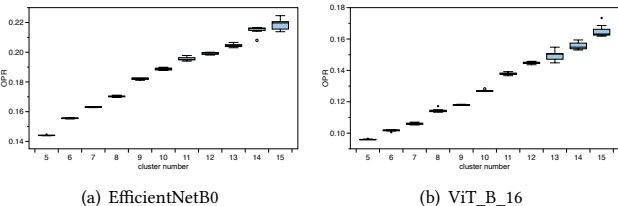

(a) EfficientNetB0

(b) ViT_B_16

**Figure 6: The OPR in the CIFAR-10 dataset when using EfficientNetB0 or ViT_b_16 as the model's backbone. A higher OPR indicates a greater occurrence of offset positive samples.**

Notice that for $r = 1, 2..., b$, we have $(h^s_{ir} h^s_{jr})^2 = 1$, $(h^t_{ir} h^t_{jr})^2 = 1$, then we get:

$$L^{ij}_{sp} = 2b - 2(h^s_{i1} h^s_{j1} h^t_{i1} h^t_{j1} + ... + h^s_{ib} h^s_{jb} h^t_{ib} h^t_{jb})$$
$$+ 2 \sum_{k \neq r; k, r = 1, ..., b} (h^s_{ik} h^s_{jk} - h^t_{ik} h^t_{jk})(h^s_{ir} h^s_{jr} - h^t_{ir} h^t_{jr}). \quad (26)$$

To minimize $L^{ij}_{sp}$, we need to maximize the second term and minimize the third term. We observe that if any $k = 1, .., b$ satisfies the following condition:

$$h^s_{ik} h^s_{jk} = h^t_{ik} h^t_{jk}, \quad (27)$$

then $L^{ij}_{sp}$ can be minimized to 0 because the second term is $2b$ and the third term is 0. Based on Eq. (27), we identify two optimization directions: (1) for any $i = 1, .., n$ and $k = 1, .., b$, $h^s_{ik} = h^t_{ik}$, which make the student model learn the individual-space knowledge from teacher model to achieve semantic space alignment; (2) there exist $i = 1, .., n$ and $k = 1, .., b$ such that $h^s_{ik} \neq h^t_{ik}$, which can not ensure semantic space alignment. That is why the SP method on the ASHP works in some cases but not in others. The SP method works well in some cases where the model tends to achieve direction (1) more, enabling the student model to achieve semantic space alignment with the teacher model. Conversely, if the optimization tends to achieve direction (2) more, it cannot achieve semantic space alignment and cannot to ensure that the hash codes from different models for one image are mapped to a close position in Hamming space.

## C THE TRAINING ALGORITHM OF BRCD

In this section, we present the training algorithm of our proposed BRCD in Algorithm 1. This method can be used in both the ASHP and the SSHP.

## D ANALYSIS OF OFFSET POSITIVE SAMPLE

In this experiment, we explored the occurrence probability of offset positive samples in CIFAR-10 when using EfficientNetB0 or ViT_B_16 as the CIBHash model's backbone. We inputted the training images $X = \{x_1, x_2, ..., x_N\}$ into the trained teacher model to obtain hash codes $H^{all}_t = \{h^t_1, h^t_2, ..., h^t_N\}$ and conducted k-means clustering on the $H^{all}_t$ to assign pseudo label $y_i$ for each image. We then created augmented images $X' = \{x_{1'}, x_{2'}, ..., x_{N'}\}$ using the same augmentation techniques as in CIBHash [34]. By inputting $X'$ into the same model, we assigned the centroid as the pseudo

---

**Algorithm 1** The training algorithm of BRCD

**Input:** a trained teacher model $f_t$, training samples $X = \{x_1, x_2, ...x_N\}$, number of cluster $k$ and the hyper-parameters $\alpha$ and $\delta$.

1: Initialization: the parameter of student model $f_s$.
2: $H^{all}_t = f_t(x \sim X)$
3: $\{Centroid_1, Centroid_2, ..., Centroid_k\}$, $\{y_1, y_2, ..., y_N\}$ = k-means($H^{all}_t$) $\{Centroid_i$ means the centroid of cluster $C_i\}$
4: calculate the bit matrix $\{e^m_1, e^m_2, ..., e^m_k\}$ according to Eqs. (8) and (9)
5: **repeat**
6:     draw a mini-batch $B = \{x_1, x_2, ..., x_M\}$ from $X$
7:     apply data augmentation on $B$ to create augmented images $B' = \{x_{1'}, x_{2'}, ..., x_{M'}\}$
8:     **for** each $i \in \{1, 2, ...., M\}$ **do**
9:         $h_{i'} = y_t(x_{i'})$
10:         set $y_{i'}$ to $x_{i'}$ according to the distance of $h_{i'}$ and $\{Centroid_1, Centroid_2, ..., Centroid_k\}$
11:     **end for**
12:     **for** each $i \in \{1, 2, ...., M\}$ **do**
13:         $R'(i) \leftarrow \{1, 2, ..., M, 1', 2'..., M'\}$
14:         $N(i) \leftarrow R'(i) - \{i, i'\}$
15:         **if** $y_i = y_{i'}$ **then**
16:             $\alpha_{i'} \leftarrow \alpha$
17:         **else**
18:             $\alpha_{i'} \leftarrow 1$
19:         **end if**
20:         **for** each $k \in N(i)$ **do**
21:             **if** $y_i = y_k$ **then**
22:                 $R'(i) \leftarrow R'(i) - \{k\}$
23:             **end if**
24:         **end for**
25:     **end for**
26:     $\hat{R}(i) \leftarrow R'(i) - \{i\}$
27:     update parameters of the student model by minimizing Eq. (10) and the loss of the semantic hashing model itself
28: **until** converged

**Output:** parameters of the student model $f_s$

---

label $y_{i'}$ for each augmentation $x_{i'}$ based on the Hamming distance. We defined the offset positive rate (OPR) as:

$$OPR = \frac{\sum_{i=1,2,...,N} I(y_i \neq y_{i'})}{N}, \quad (28)$$

where $I$ is the indicator function. A higher OPR indicates a greater occurrence of offset positive samples. We set various cluster numbers and conducted repeated experiments to increase the credibility of the results. The box plots in Figure 6 show that the OPR is significant, and with an increasing number of clusters, the OPR also increases. This outcome demonstrates that regardless of whether a lightweight backbone EfficientNetB0 or a more powerful model ViT_B_16 is employed, the "offset positive sample" is present. Besides, these findings support our consideration that removing offset positive samples is necessary for effective knowledge distillation. Intuitively, reducing the degree of image augmentation can decrease OPR, but this may also reduce the number of hard positive

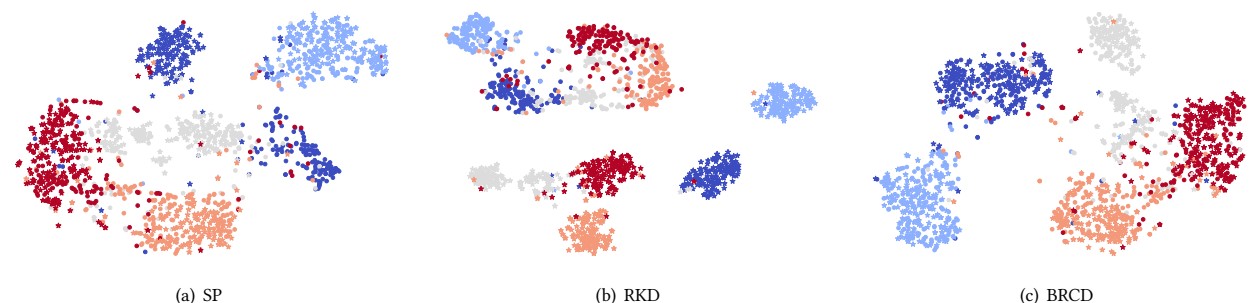

(a) SP                                                    (b) RKD                                                    (c) BRCD

**Figure 7: Visualization of the teacher and student models' hash codes on the CIFAR-10 dataset.**

**Table 3: The average inference time for different batches of image data on various backbones (ms).**

| Backbone/Settings \ Batch Size | 4 | 8 | 16 | 32 | 64 | 128 | 256 | 512 | Params | GFlops |
|---|---|---|---|---|---|---|---|---|---|---|
| ViT_B_16 | 14.32 | 27.21 | 52.35 | 104.16 | 205.53 | 415.55 | 814.24 | 1796.03 | 86.6M | 17.56 |
| ViT_L_16 | 46.34 | 88.61 | 169.04 | 344.96 | 657.67 | 1354.89 | 2673.72 | 5454.58 | 304.3M | 61.55 |
| ResNet18 | 6.58 | 10.48 | 19.58 | 35.83 | 68.22 | 164.84 | 333.79 | 648.53 | 11.7M | 1.81 |
| MobileNetV2 | 9.35 | 12.98 | 20.85 | 36.91 | 69.67 | 162.54 | 323.27 | 642.99 | 3.5M | 0.3 |
| EfficientNetB0 | 12.75 | 15.81 | 24.36 | 40.98 | 89.20 | 175.33 | 313.07 | 637.89 | 5.3M | 0.39 |
| n=10,000,000; k=100 | 2.47 | 2.61 | 2.75 | 2.85 | 4.70 | 6.67 | 12.16 | 24.13 | - | - |
| n=10,000,000; k=1000 | 5.25 | 5.87 | 7.97 | 8.46 | 13.38 | 17.36 | 28.47 | 65.26 | - | - |
| n=30,000,000; k=100 | 5.52 | 7.59 | 10.11 | 13.39 | 14.54 | 22.28 | 36.68 | 65.61 | - | - |
| n=30,000,000; k=1000 | 12.38 | 18.54 | 26.64 | 37.84 | 45.36 | 68.16 | 106.63 | 174.84 | - | - |
| n=50,000,000; k=100 | 8.96 | 11.7 | 14.02 | 16.17 | 20.7 | 32.91 | 61.51 | 109.77 | - | - |
| n=50,000,000; k=1000 | 20.15 | 27.34 | 37.63 | 47.73 | 61.36 | 99.62 | 196.35 | 347.72 | - | - |

samples in the augmentations, leading to performance degradation. Therefore, there could be a trade-off in this process, which requires further research in the future.

## E   VISUALIZATION ANALYSIS

To assess whether the learned semantic distribution from the student model is aligned with the teacher model from a visual perspective, we plot the embedding of images in the Hamming space using 2-D t-SNE projection. Specifically, we compare the BRCD method with the SP and RKD methods, using a bit length of 32 on the CIFAR-10 dataset. Figure 7 showcases this comparison. Each point in the plot corresponds to an image, with images of the same class being depicted in the same color. To ensure clarity, we have chosen to display the first five categories. In this visualization, the color red represents airplanes, orange represents automobiles, gray represents birds, blue represents cats, and dark blue represents deer. Circles represent hash codes generated by the student model, while pentagrams represent hash codes generated by the teacher model.

Figure 7 (c) shows that our proposed BRCD method makes hash codes from different models with the same class close in the space. In contrast, as illustrated in Figure 7 (b), RKD can preserve the relation between student and teacher model images but cannot ensure semantic space alignment. On the other hand, as shown in Figure 7 (a), the SP method produces distinct distillation outcomes for different categories. Except for the dark blue category (deer),

which does not align, the remaining categories exhibit alignment, further validating the conclusion drawn in section B. This visualization analysis serves as an intuitive verification of the effectiveness of our approach in achieving semantic space alignment.

## F   INFERENCE AND SEARCH TIME ANALYSIS

This section presents an analysis of the time required for model inference and hash code search. We provide a detailed examination of key factors that significantly impact these processes. We explore different backbones and batch sizes to assess their influence on inference time. Additionally, we investigate the impact of varying candidate sizes, the number of relevant images to retrieve, and batch sizes on search time. It is worth noting that the inference and search time can also be affected by various factors, including code implementation, used libraries, software versions, and other hardware-related factors in practice. We implement our experiments on our server as described in Section 5.3. To construct the index, we utilized the faiss library [4].

Table 3 provides an overview of the inference time overhead and search time overhead. The upper part of the table shows the inference time of the CIBHash [34] with different backbones and batch sizes. Although large-scale backbones provide better performance for semantic hashing models, it is noteworthy that the inference time of ViT_L_16 is approximately seven times higher than that of

---

[4]https://github.com/facebookresearch/faiss

 

smaller models such as ResNet18. The lower part of the table shows the search time required to locate relevant images after obtaining the hash codes from the CIBHash model, where $n$ denotes the size of the candidate set, and $k$ represents the retrieval of the top $k$ most relevant images. As expected, a larger candidate set size and a larger value of $k$ will take more time to find relevant images. Moreover, we can find that the search time overhead is higher than the inference time when considering the same batch size. When using a large batch size, the inference time becomes the dominant factor in the overall time consumption of the retrieval procedure.