# OpenReview forum: "Bit-mask Robust Contrastive Knowledge Distillation for Unsupervised Semantic Hashing"
_ACM.org/TheWebConf/2024/Conference — TheWebConf24_

### Official Review · Reviewer_xU7T · 2023-11-14

**Novelty:** 6
**Technical Quality:** 6

**Review:**

In knowledge distillation a so called “student” model is a lightweight model for image feature extraction that is trained to mimic a more complex and heavy “teacher” model. The author of the paper “Bit-mask Robust Contrastive Knowledge Distillation for Unsupervised Semantic Hashing” propose to apply semantic hashing in a Knowledge Destillation context and provide optimisation approaches for their BRCD model. To this end they first optimise the model student to mimic the teacher model and at the same time optimise the hash function suich taht semantically similar objects will be closer in hamming space. The authors perform a study, where they compare the performance of top-tire image hashing algorithms and show that their semantic hashing approach outperform the state of the art.
The paper is clearly written and a pleasure to read. The idea is nice and the approach is credible.


The paper titled "Bit-mask Robust Contrastive Knowledge Distillation for Unsupervised Semantic Hashing" explores the concept of knowledge distillation, where a simpler, more efficient "student" model is trained to replicate the functionality of a larger, more complex "teacher" model, specifically for image feature extraction. The novel approach presented in the paper extends the approach with semantic hashing. The authors introduce the BRCD (Bit-mask Robust Contrastive Distillation) model, which focuses on optimizing the student model to not only imitate the teacher model, but also to refine the hash function, such that semantically similar images are positioned closer together in the Hamming space.

**Questions:**

The authors provide an empirical study comparing the BRCD model with leading image hashing algorithms. The results indicate that the proposed semantic hashing method outperforms current state-of-the-art techniques in performance. The paper is commended for its clarity and readability, presenting a credible and innovative approach in the field of unsupervised semantic hashing.

Could you, please, elaborate on the significance of the results presented in Table 1. Even if you have information in Appendix, usually it can not be considered and have to be in the paper body.

Choice of mAP@1000 as a Metric: What is the rationale behind using mAP@1000 as a key performance metric, as opposed to a smaller scale like mAP@10? How does this choice affect the interpretation of the model's effectiveness in semantic hashing?

Please, explain, why were specific datasets, such as CIFAR-10, chosen for this study? Given that CIFAR-10 is considered a relatively simple dataset, how does BRCD perform with respect of the complexity of the dataset contents?

**Ethics Review Description:**

No ethical issues found.

**Reviewer Confidence:**

2: The reviewer is willing to defend the evaluation, but it is likely that the reviewer did not understand parts of the paper

**Scope:**

4: The work is relevant to the Web and to the track, and is of broad interest to the community

---

### Official Review · Reviewer_XZTH · 2023-11-15

**Novelty:** 4
**Technical Quality:** 5

**Review:**

Motivated by the observation that the prevailing knowledge distillation approaches ignore the unique search paradigm of semantic hashing, the inherent necessities of the distillation
process, and the property of hash codes, this paper proposed an innovative Bit-mask Robust Contrastive knowledge Distillation (BRCD) method, specifically devised for the distillation of semantic hashing models.
Overall, the idea is interesting, the work is well written and easy to follow.

The pros and cons are as follows:

- P-1: pinpoint the shortcomings of previous methods in ignoring the unique search paradigm of semantic hashing, the inherent necessities of the distillation process, and the property of hash codes, a new framework is proposed to effectively cope with these issues;
- P-2: the work is well written and easy to follow;
- P-3: a series of experiments are conducted over multiple datasets in order to demonstrate the effectiveness of the proposed framework.

- C-1: the experimental analysis is not sufficient. For example, (1) significance test is necessary to clearly show the performance difference, such as between KL and the BRCD; (2) in the context of image retrieval, the result comparisons with smaller cutoff values, say @K=10, 50, 100, are also important due to the position/click bias.
- C-2: it is not clear why CIFAR-10 is used for comparisons after section 5.5.
- C-3: the source code is not provided, which may impact the reproducibility.

**Questions:**

Q1: it is not clear why CIFAR-10 is used for comparisons after section 5.5

**Ethics Review Description:**

TBA

**Reviewer Confidence:**

3: The reviewer is confident but not certain that the evaluation is correct

**Scope:**

4: The work is relevant to the Web and to the track, and is of broad interest to the community

---

### Official Review · Reviewer_oXeb · 2023-11-25

**Novelty:** 6
**Technical Quality:** 6

**Review:**

This work observes the limitations of current knowledge distillation methods for semantic hashing, based on which proposes a new treatment of bit-mask robust contrastive knowledge distillation that applies to both symmetric and asymmetric semantic hashing paradigms. Specifically, two contrastive objectives are introduced to align the semantic space with the student model. A clustering method is employed to detect and discard the augmented sample that might mislead the optimization. Besides, redundant bits are reduced for efficiency concerns. Experiments show a performance boost of the proposed method. The manuscript is well presented with clear motivation and interesting observations.

**Questions:**

- I’m not sure what the effect of the proposed clustering method is. Does it identify positive samples that might mislead the optimization? Is it possible to provide some empirical evidence about this part?
- How many (e.g., in percentage) augmented samples can be categorized as the “off positive samples” that might mislead the optimization? The benefit of using augmented positive data will be limited if the number is large. Does the structural-semantic KD and individual-space KD employ the off-positive samples? It seems that off-positive samples take effect in both terms, according to Fig. 2.

**Reviewer Confidence:**

3: The reviewer is confident but not certain that the evaluation is correct

**Scope:**

4: The work is relevant to the Web and to the track, and is of broad interest to the community

---

### Official Review · Reviewer_EkDc · 2023-11-29

**Novelty:** 6
**Technical Quality:** 5

**Review:**

In this paper, the authors propose a new contrastive knowledge distillation method and verify its effectiveness on the unsupervised semantic hashing image retrieval task. Specifically, the authors target at achieving the semantic space alignment and employ a cluster-based method to solve the offset positive sample problem. Extensive experiments are performed on three datasets to compare the performance using different KL methods and different hash models, and different backbones. The performance improvement is promising. Overall, the writing is good and easy to follow.
Cons:
(1)In section 4.1, the concerns that minimizing the distances of the output of teacher and student model regarding the same sample, and minimizing the similar  image pairs and maximizing the dissimilar images pairs are universal both in the KL field and the hashing retrieval field, what is the novelty of the proposed method here?
(2)In section 4.2, the authors first perform the k-means clustering on the set of hash codes of all training images and then assign pseudo labels to image based on the closest centroid. This operation is important in the proposed method. However, in the unsupervised scenario, the value of k is important, especially when the number of labels are unknown. But in the paper, the authors overlook the analysis of k for different datasets. Some important details regarding k should be added and illustrated.
(3)In section 5.7.23, the analysis of alpha and sigma are not enough. The authors should list the best value of these two parameters for different datasets. It seems that their values should be different for different dataset.

**Questions:**

(1)Compared with MSCOCO and ImageNet100, the CIFAR-10 dataset is very small. However, except the table 1, the experimental results and analysis are only related to this small dataset, which is insufficient. More analysis should be done on other two datasets to validate the generality effectiveness of the proposed method.
(2)What is the relations between equation 9 and 10?  How the authors use the obtained bit mask in the following loss function of equation 10? More clarification should be introduced for this problem. Current, the introduction is unclear.
(3)What is the effect of the clustering results to the proposed method? It seems that the clustering process is important to remove the false negative and offset positive samples. Thus, some key experiments are overlooked to better verify the effectiveness of the proposed methos.

**Ethics Review Description:**

Noethical issues

**Reviewer Confidence:**

4: The reviewer is certain that the evaluation is correct and very familiar with the relevant literature

**Scope:**

4: The work is relevant to the Web and to the track, and is of broad interest to the community

---

### Official Review · Reviewer_NSDL · 2023-11-30

**Novelty:** 4
**Technical Quality:** 5

**Review:**

This paper focuses on a significant problem: how can latency for semantic hashing search models be reduced? It proposes a knowledge distillation framework to build a student model that can make quick inferences. The knowledge is transferred from a powerful teacher model to the student model via unsupervised learning tasks.

Strengths: Knowledge distillation (KD) is the de-facto approach for online latency issues in many tasks. Rather than simply focusing on alignment between teacher and student, this paper proposes structural-semantic knowledge with robust optimization. Moreover, after distillation, the paper finds redundancy bit based on the bit independence assumption. The experimental results show the improvement of the proposed framework over other KD methods. And the ablation study supports the modeling choices.

Weaknesses: The semantic space alignment task should include structural-semantic knowledge if the teacher model is powerful enough. Basically, Eq. 1 should include Eq. 2, if enough data for distillation is available. Moreover, I agree with the robustness issue when employing data augmentation. However, this issue should be addressed in the data augmentation step, not the distillation step. I don't see why dealing offset positives in the loss function.

**Questions:**

1. What's the performance if only distill based on Eq. 1?

2. What's the difference between listed baselines and BRCD? I think more detail is needed to understand Table 1 completely.

3. Why are some of the listed baselines not working in the ASHP paradigm?

**Reviewer Confidence:**

2: The reviewer is willing to defend the evaluation, but it is likely that the reviewer did not understand parts of the paper

**Scope:**

4: The work is relevant to the Web and to the track, and is of broad interest to the community

---

### Decision · Program_Chairs · 2024-01-22

**Decision:**

Accept

**Comment:**

This paper identifies the limitations of existing knowledge distillation techniques for semantic hashing and proposes an approach applicable to both symmetric and asymmetric semantic hashing paradigms. The reviewers praised the novelty and readability of the paper as well as the motivation of the proposed method. The experiments were convincing in general. Some reviewers raised questions about the experiments and unclear introduction, and they adjusted their scores based on the authors' responses. Based on the reviews, rebuttal, and my own reading, I recommend acceptance of the paper. I suggest the authors incorporate the feedback from the reviewers in the revised paper.